# Bioinformatics Analysis of Gene Expression Profiles for Diagnosing Sepsis and Risk Prediction in Patients with Sepsis

**DOI:** 10.3390/ijms24119362

**Published:** 2023-05-27

**Authors:** Hayoung Choi, Jin Young Lee, Hongseok Yoo, Kyeongman Jeon

**Affiliations:** 1Division of Pulmonary, Allergy, and Critical Care Medicine, Department of Internal Medicine, Hallym University Kangnam Sacred Heart Hospital, Hallym University College of Medicine, Seoul 07441, Republic of Korea; hychoimd@gmail.com; 2Division of Pulmonary and Critical Care Medicine, Department of Medicine, Samsung Medical Center, Sungkyunkwan University School of Medicine, Seoul 06351, Republic of Korea; yenayein@gmail.com (J.Y.L.); hongseok.yoo@samsung.com (H.Y.); 3Department of Health Sciences and Technology, SAIHST, Sungkyunkawan University, Seoul 06351, Republic of Korea

**Keywords:** sepsis, biomarkers, diagnosis, genes, bioinformatics

## Abstract

Although early recognition of sepsis is essential for timely treatment and can improve sepsis outcomes, no marker has demonstrated sufficient discriminatory power to diagnose sepsis. This study aimed to compare gene expression profiles between patients with sepsis and healthy volunteers to determine the accuracy of these profiles in diagnosing sepsis and to predict sepsis outcomes by combining bioinformatics data with molecular experiments and clinical information. We identified 422 differentially expressed genes (DEGs) between the sepsis and control groups, of which 93 immune-related DEGs were considered for further studies due to immune-related pathways being the most highly enriched. Key genes upregulated during sepsis, including S100A8, S100A9, and CR1, are responsible for cell cycle regulation and immune responses. Key downregulated genes, including CD79A, HLA-DQB2, PLD4, and CCR7, are responsible for immune responses. Furthermore, the key upregulated genes showed excellent to fair accuracy in diagnosing sepsis (area under the curve 0.747–0.931) and predicting in-hospital mortality (0.863–0.966) of patients with sepsis. In contrast, the key downregulated genes showed excellent accuracy in predicting mortality of patients with sepsis (0.918–0.961) but failed to effectively diagnosis sepsis. In conclusion, bioinformatics analysis identified key genes that may serve as biomarkers for diagnosing sepsis and predicting outcomes among patients with sepsis.

## 1. Introduction

Sepsis is a life-threatening organ dysfunction caused by a dysregulated host response [1]. Sepsis and septic shock are major healthcare problems that affect millions of patients worldwide each year and kill approximately 17–33% of those affected [2,3]. Unfortunately, the reported incidence of sepsis and its associated healthcare burden are both currently on the rise due to the global trend of population aging and patients having a greater number of comorbidities [4,5]. Biomarker development could serve as a cornerstone of sepsis management and may ameliorate the healthcare burden of sepsis because early recognition is essential for timely treatment and the improvement of sepsis outcomes [6].

Numerous biomarkers for sepsis, including C-reactive protein and procalcitonin, have been investigated previously [7,8]. However, to date no marker has demonstrated sufficient discriminatory power [9,10]. Bioinformatics approaches integrate computational and life sciences to screen molecular and clinical data via data mining, pathway analysis, statistical analysis, and visual processing. These methods can investigate disease on the molecular level and have been widely used to identify significant biomarkers for sepsis [11,12,13,14,15]. In addition to the previous studies using datasets downloaded from public repositories, some prospective cohort studies also performed bioinformatics analysis to explore potential biomarkers for sepsis diagnosis [16,17]. Although previous studies have provided important insight into biomarkers and the pathophysiology of sepsis, more bioinformatics studies are warranted to validate the study results in association with real-world clinical outcomes.

Thus, the present study aimed to (1) use bioinformatics analyses to explore key differentially expressed genes (DEGs) between patients with sepsis from a prospective cohort and healthy volunteers and (2) validate the accuracy of bioinformatics analyses for diagnosing sepsis and predicting sepsis outcomes by integrating molecular experiments with clinical information.

## 2. Results

### 2.1. Clinical Characteristics of Patients with Sepsis

During the study period, we enrolled 133 critically ill patients with sepsis after excluding 63 who were not diagnosed with sepsis and two who withdrew their consent. Of the 133 patients, 90 (67.7%) were male, and the median age of patients was 66 years (interquartile range (IQR), 58–73 years). When patients were admitted to intensive care units, 60 (45.1%) and 69 (51.9%) received mechanical ventilation therapy and vasopressors, respectively. With respect to the severity of illness, the median SAPS 3 score was 55 (IQR, 47–63), the APACHE II score was 24 (IQR, 20–30), and the initial Sequential Organ Failure Assessment score was 9 (IQR, 7–11). The rates of 28-day mortality and in-hospital mortality were 16.5% and 24.8%, respectively (Table 1).

### 2.2. Identification of Candidate mRNAs: Bioinformatics Analyses

This study initially identified 422 DEGs between 133 patients with sepsis and 12 healthy volunteers. A principal component analysis (PCA) of global gene expression profiles revealed that sepsis patients were clearly separate from healthy volunteers (Figure 1A). In addition, distinct patterns of gene expression existed in sepsis patients when transcriptomic profiles of sepsis patients were compared to those of healthy volunteers (Figure 1B). Further bioinformatics analyses were conducted to identify key genes related to sepsis. First, enriched gene ontology (GO) functional analysis revealed that the identified DEGs were mainly involved in the immune response (Figure 1C,D). 

Second, a protein–protein interaction (PPI) network analysis of the 422 DEGs also showed that the most extensive module was composed of 78 seeds, 1381 nodes, and 1823 edges. Moreover, it appeared to be most strongly enriched in immune-related pathways (Figure 2). 

A following PPI network analysis of 93 immune-related DEGs revealed that these enriched immune-related pathways included adaptive immune response, positive regulation of immune response, positive regulation of leukocyte cell–cell adhesion, cell activation, and positive regulation of cytokine production (Figure 3).

Results of a molecular complex detection (MCODE) analysis performed to screen the significant modules of the PPI network are shown in Figure 4.

The 93 immune-related DEGs identified here included 51 upregulated and 42 downregulated genes in patients with sepsis compared to healthy volunteers. Table 2 summarizes the top 10 upregulated and downregulated genes in patients with sepsis. Significantly upregulated genes included S100A8, VNN1, HMGB2, and S100A9, whereas significantly downregulated genes included CD79A, HLA-DQB2, PLD4, and CCR7.

### 2.3. Experimental and Clinical Validation of Potential Biomarkers from the mRNA Profile

Next, we used quantitative real-time PCR (qPCR) to measure the expression levels of identified DEGs and thereby validate our bioinformatics analyses using molecular data in 133 patients with sepsis and 12 healthy volunteers. Among the 10 upregulated and 10 downregulated genes, 3 upregulated and 4 downregulated genes showed consistent results with bioinformatics analysis. The three upregulated genes, S100A8, S100A9, and CR1, showed significantly higher expression levels in patients with sepsis than in healthy volunteers (*p* < 0.001 for S100A8 and S100A9, and *p* = 0.005 for CR1), whereas the four downregulated genes, CD79A, HLA-DQB2, PLD4, and CCR7, showed significantly lower expression levels in patients with sepsis than in healthy volunteers (*p* < 0.001 for all four genes) (Table 3).

A receiver operating characteristic (ROC) curve analysis was then conducted to determine the accuracy of these seven genes in diagnosing sepsis: S100A8 showed very high accuracy (i.e., area under the curve (AUC): 0.931, 95% confidence interval (CI): 0.880–0.982), while both S100A9 (AUC: 0.791, 95% CI: 0.711–0.871) and CR1 (AUC: 0.747, 95% CI: 0.647–0.820) showed fair accuracy. In contrast to the three upregulated genes, all four downregulated genes failed to effectively discriminate between patients with sepsis and healthy volunteers (Table 4).

Further analyses were then performed to compare gene expression in patients with sepsis who died and survived among the 133 with sepsis. All three upregulated genes (S100A8, S100A9, and CR1) and all four downregulated genes (CD79A, HLA-DQB2, PLD4, and CCR7) showed significantly higher expression levels in the dead than in the survivors among patients with sepsis (*p* < 0.001 for all) (Table 3). The ROC curve analysis was then used to assess the accuracy of the seven genes in predicting in-hospital mortality among patients with sepsis. All upregulated and downregulated genes demonstrated excellent accuracy, with CR1 showing the highest accuracy (AUC: 0.966, 95% CI: 0.939–0.993), followed by CD79A (AUC: 0.961, 95% CI: 0.930–0.992) and HLA-DQB2 (AUC: 0.936, 95% CI: 0.895–0.978) (Table 4).

## 3. Discussion

In the present study, bioinformatics analysis revealed that immune-related pathways were strongly enriched in patients with sepsis relative to healthy volunteers. In addition, we identified three key genes that were upregulated in patients with sepsis (S100A8, S100A9, and CR1) as well as four key genes that were downregulated (CD79A, HLA-DQB2, PLD4, and CCR7). Furthermore, validation of these findings by molecular experiments and clinical outcomes determined that the key upregulated genes showed excellent to fair accuracy for both diagnosing sepsis and predicting in-hospital mortality of patients with sepsis. In contrast, the key downregulated genes showed excellent accuracy in predicting in-hospital mortality of patients with sepsis but failed to effectively diagnose sepsis.

Immune-related pathways were the most highly enriched biological pathways in patients with sepsis relative to healthy volunteers; this finding is consistent with previous bioinformatics studies, which also revealed DEGs were significantly enriched in pathways related to neutrophil activation, the TNF signaling pathway, and cytokine secretion [11,13,14,15,16,18]. These results suggest that the pathophysiology of sepsis is driven by “an aberrant or dysregulated host response to infection”, which is also reflected in the current definition of sepsis [1]. Furthermore, a recent RNA sequencing study provided more insights into the pathophysiology of sepsis. The study compared RNA sequencing results between 18 immunocompromised patients with sepsis and 18 Sequential Organ Failure Assessment score-matched immunocompetent controls, and it demonstrated that patients with sepsis were more likely to show compromised T cell function, decreased T cell diversity, and altered metabolic signaling than controls [19]. In this way, bioinformatics studies will increasingly contribute to unveiling the pathophysiology of sepsis.

Notably, the key genes upregulated during sepsis are responsible for regulating cell cycle progression and differentiation (i.e., S100A8 and S100A9) and immune responses (i.e., ANXA1, APOBEC3A, LILRA5, CR1, and CD55) [20]. In agreement with our findings, one Chinese prospective cohort study also performed a comprehensive transcriptome profile analysis and qPCR validation, and suggested S100A8, S100A9, and ANXA3 as key genes differentially expressed between sepsis patients and healthy controls [16]. Potential underlying mechanisms of upregulated S100A8 and S100A9 may also include altering MyD88-dependent gene programs, which consequently prevent hyperinflammatory responses without impairing pathogen defense [21], and mediating endotoxin-induced cardiomyocyte dysfunction [16,22]. Our qPCR results and ROC curve analyses confirmed the accuracy of S100A8, S100A9, and CR1 in diagnosing sepsis as well as in predicting in-hospital mortality among patients with sepsis. Thus, our results suggest that the expression levels of the key upregulated genes identified here may serve as potential biomarkers of sepsis. 

In this study, the key genes downregulated during sepsis also encoded proteins responsible for the immune response, including the expression of antigen-presenting cells, the regulation of cytokine production, and the activation of B and T lymphocytes (i.e., CD79A, HLA-DQB2, PLD4, and CCR7) [20]. This is in line with prior studies showing that genes involving Jak-STAT signaling, T cell receptor signaling, and natural killer cell-mediated pathways were downregulated [23,24,25]. The results showed that genes participating in the immune response have mixed differential expression patterns of both up- and downregulation. Considering sepsis has been found to manifest a balance between competing pro- and anti-inflammatory pathways [26], the downregulation of immune response genes in patients with sepsis implies the homeostatic regulation of immunity during sepsis. Thus, the failed homeostatic regulation of immunity may have consequently resulted in the development and progression of sepsis. Interestingly, the key downregulated genes revealed an excellent accuracy in predicting in-hospital mortality of patients with sepsis but failed to diagnose sepsis in the ROC curve analyses. Taken together, key downregulated genes helped us understand more about sepsis pathophysiology; however, they may not be useful as sepsis biomarkers compared with key upregulated genes. 

This study’s most important strength is the validation of key DEGs between patients with sepsis and healthy volunteers via molecular experiments and clinical information from a prospective cohort. These validation methods can elucidate which genes may act as biomarkers of the diagnosis and mortality prediction of sepsis. Nonetheless, two limitations to this study should also be acknowledged. First, the study population was relatively small; thus, future bioinformatics studies, including a larger sample size, are necessary to confirm our findings. Second, this study was conducted in Korea, which might limit the generalizability of our results to other countries or ethnic groups.

## 4. Materials and Methods

### 4.1. Study Population

This study included patients with sepsis from the Samsung Medical Center Registry of Critical Illness (SMC-RoCI), a prospective observational study conducted at the Samsung Medical Center (i.e., a 1989-bed, university-affiliated, tertiary referral hospital in Seoul, Republic of Korea) between October 2015 and January 2020 as previously described [27]. Sepsis was defined according to the third edition of the International Consensus Definitions for Sepsis and Septic Shock (Sepsis-3) [1]. Consequently, patients enrolled before the release of this new definition were reclassified according to the Sepsis-3 scheme.

In addition to patients with sepsis, we used a control consisting of 12 healthy volunteers (≥19 years of age) who donated blood specimens for research purposes. Written informed consent was obtained from all participants or their legally authorized representatives before enrollment. This study was conducted according to the Declaration of Helsinki, and all experimental procedures were approved by the institutional review board of the Samsung Medical Center (Application No. 2013-12-033).

### 4.2. Sample Collection

Blood samples consisted of 19 mL of whole blood collected into ethylenediaminetetraacetic acid tubes within 48 h of enrollment in the SMC-ROCI. Samples were centrifuged at 480× *g* (Eppendorf Centrifuge 5810 No. 0012529-rotor A-4-81) for 10 min at 4 °C within 4 h of collection. Several plasma aliquots from each study participant were then isolated and stored at −80 °C for further analysis.

### 4.3. Total RNA Isolation and Quality Analysis

For RNA isolation, whole blood (2 mL) was also collected in PAXgene tubes, using BD PAXgene blood RNA tubes (BD, cat. no. 762165). Total RNA was isolated from whole blood using the TRIzol reagent (Invitrogen, Carlsbad, CA, USA) following the manufacturer’s protocol [28]. RNA quantity and purity were measured using a NanoDrop 2000 (Thermo Fisher Scientific, Wilmington, DE, USA). RNA quality, yield, and distribution were determined using an Agilent 2100 Bioanalyzer (Agilent Technologies, Santa Clara, CA, USA) [29]. Blood samples (5 mL) were also collected from healthy volunteers; these samples were also prepared using the method described above. 

### 4.4. Library Preparation and Sequencing

Libraries were prepared from total RNA using a NEBNext Ultra II Directional RNA-Seq Kit (New England BioLabs, UK). Messenger RNA (mRNA) was isolated using a Poly(A) RNA Selection Kit (Lexogen, Inc., Vienna, Austria). Isolated mRNA was then used for cDNA synthesis and shearing by following the manufacturer’s instructions. Indexing was performed using Illumina indices 1–12. Enrichment was performed by polymerase chain reaction (PCR). Subsequently, libraries were checked using an Agilent 2100 Bioanalyzer (DNA High Sensitivity Kit) to evaluate the mean fragment size. Quantification was performed using a library quantification kit on a StepOne Real-Time PCR System (Applied Biosystems Life Technologies, Carlsbad, CA, USA). High-throughput sequencing was performed as paired-end 100 bp sequencing on a NovaSeq 6000 sequencing platform (Illumina, Inc., San Diego, CA, USA) [29,30].

### 4.5. Data Analysis

Quality control of the raw sequencing data was performed using FastQC [31]. Adapter sequences and low-quality reads (<Q20) were removed using the fastx_clipper function implemented in the FASTX_Toolkit and by BBMap [32]. Trimmed reads were then mapped to the reference genome using TopHat [33]. Gene expression levels were estimated as fragments per kilobase of transcript per million (FPKM) mapped reads values as determined by Cufflinks [34]. All FPKM values were normalized based on the quantile normalization method implemented by the EdgeR package for R [35].

### 4.6. Identification of DEGs

GEO2R was used to screen for DEGs between the sepsis and control groups. GEO2R is an R-based interactive web tool that helps to identify and visualize differential gene expression [36]. PCA of the different groups’ samples was performed on the gene expression matrix. We set the threshold of differential expression to the default standard (i.e., |log2 (fold change [FC])| > 1 and adjusted *p* < 0.05) to identify significant DEGs between the two groups. Thus, significantly upregulated DEGs showed log2 FC > 1, and significantly downregulated DEGs showed log2 FC < 1 [37]. Significance was defined as an adjusted *p* value < 0.05 to control for type I errors in multiple tests.

### 4.7. Functional and Pathway Enrichment Analyses

To further recognize the underlying biological functions of the DEGs identified in the previous step, we performed GO functional analysis to annotate all DEGs according to the three main GO categories: molecular function, cellular component, and biological process [38]. Furthermore, to further elucidate the DEG pathways, we performed a Kyoto Encyclopedia of Genes and Genomes (KEGG) pathway enrichment analysis [39]. For GO functional annotation and KEGG pathway enrichment analyses, we used the WEB-based Gene SeT AnaLysis Toolkit (WebGestalt), the web-based Database for Annotation, Visualization, and Integrated Discovery (DAVID) tool version 6.8 [37,40], and Metascape [41].

### 4.8. Protein–Protein Interaction Network Analysis

Because proteins function in a coordinated manner within a complicated and dynamic network, we constructed a PPI network of the target genes using the Search Tool for the Retrieval of Interacting Genes (STRING) database, version 11.0 [42]. In addition, the Cytoscape plug-in Network Analyzer was also used for further analyses. Furthermore, the topological properties of the PPI network, including node degree, were calculated by searching for hub genes using the PPI network [43]. MCODE analysis implemented in Cytoscape was then performed to screen for significant modules of the PPI network using the following cut-off parameters: node score cut-off = 0.2, K-core = 2, and degree cut-off = 2.

### 4.9. Quantitative Real-Time PCR

To evaluate the expression levels of key genes, we performed qPCR analyses in duplicate. The reaction conditions were as follows: an initial step of 50 °C for 2 min, denaturing at 95 °C for 5 min, followed by 40 cycles of 95 °C for 30 s and 58.5 °C for 1 min. qPCR was performed on an ABI ViiA 7 Real-Time PCR System (Applied Biosystems) and was followed by a melting curve analysis. Glyceraldehyde-3-phosphate dehydrogenase was selected as an internal control. The 2^−ΔΔ*CT*^ algorithm (ΔCT = Ct. target − Ct. reference) was employed for downstream data analysis [44].

### 4.10. Statistical Analysis

Categorical variables were compared using the chi-square or Fisher’s exact tests. Continuous variables were compared using Mann–Whitney *U* tests. For clinical validation of bioinformatics analysis results, an ROC curve was used to analyze the diagnostic accuracy of mRNA expression for (1) discriminating between patients with sepsis and healthy volunteers and (2) predicting in-hospital mortality among patients with sepsis. The sensitivity and specificity were also calculated to suggest the optimal cut-off value of each gene. All tests were two-tailed, and *p* < 0.05 was used as the threshold of statistical significance. Data were analyzed using STATA version 16 (Stata Corp., College Station, TX, USA).

## 5. Conclusions

Bioinformatics analysis revealed that immune-related pathways were the most enriched in patients with sepsis relative to healthy volunteers. In addition, we identified key genes that were upregulated in sepsis, namely, S100A8, S100A9, and CR1, as well as those that were downregulated, namely, CD79A, HLA-DQB2, PLD4, and CCR7. The key upregulated genes showed excellent to fair accuracy in diagnosing sepsis and predicting in-hospital sepsis mortality; however, the key downregulated genes showed excellent accuracy in predicting in-hospital sepsis mortality but failed to effectively diagnose sepsis.

## Figures and Tables

**Figure 1 ijms-24-09362-f001:**
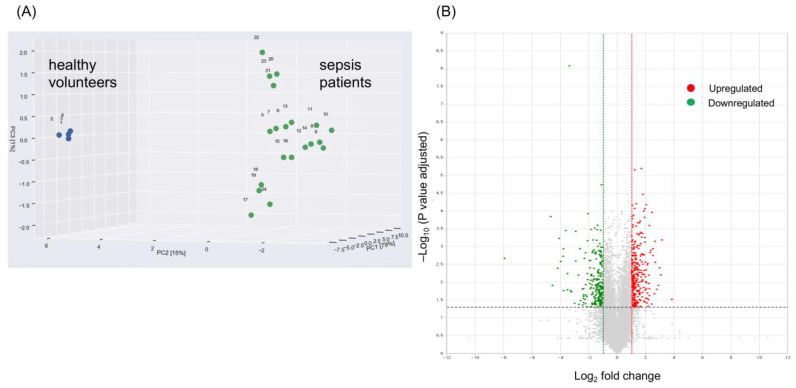
Gene ontology (GO) enrichment analysis of 422 differentially expressed genes (DEGs) between patients with sepsis and healthy volunteers. (**A**) Principal component analysis of the RNA transcriptome from sepsis patients (green) and that from healthy volunteers (blue). (**B**) Volcano plot of DEGs showing upregulated genes in red and downregulated genes in green. (**C**) GO enrichment analysis visualizing main DEGs that are mainly involved in the response to sepsis. (**D**) Number of identified DEGs according to their biological process (red), cellular component (blue), or molecular function (green) categorization.

**Figure 2 ijms-24-09362-f002:**
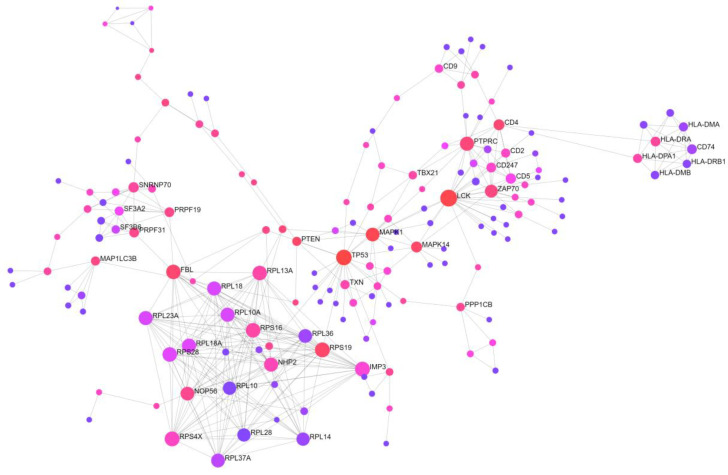
Protein–protein interaction network analysis of all 422 differentially expressed genes between patients with sepsis and healthy volunteers. Immune-related pathways, presented as the largest module (78 seeds, 1381 nodes, and 1823 edges), show the greatest enrichment.

**Figure 3 ijms-24-09362-f003:**
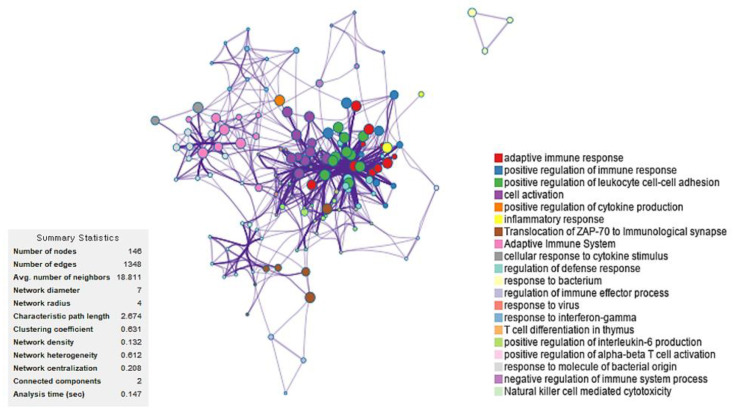
Protein–protein interaction network analysis of 93 immune-related differentially expressed genes between patients with sepsis and healthy volunteers. Enriched pathways are also shown.

**Figure 4 ijms-24-09362-f004:**
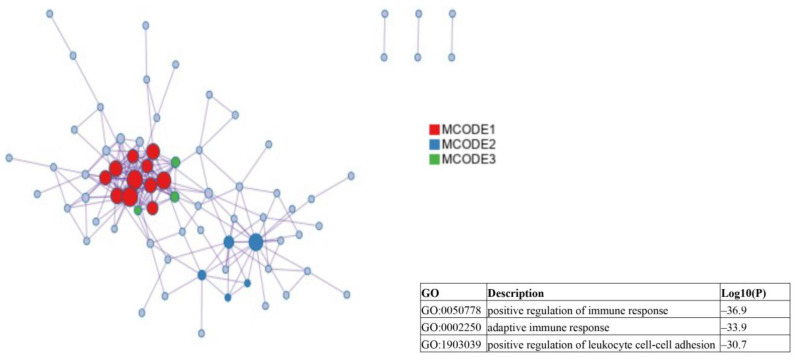
Molecular complex detection analysis results used to screen the significant modules identified in the protein–protein interaction network.

**Table 1 ijms-24-09362-t001:** Characteristics of patients with sepsis.

Variable	Value (n = 133)
Age (years)	66 (58–73)
Male	90 (67.7)
Comorbidities	
All malignancies	51 (38.3)
Solid organ malignancies	35 (26.3)
Hematologic malignancies	16 (12.0)
Diabetes mellitus	41 (30.8)
Chronic obstructive pulmonary disease	16 (12.0)
Chronic kidney disease	10 (7.5)
Myocardial infarction	8 (6.0)
Congestive heart failure	7 (5.3)
Cerebrovascular disease	8 (6.0)
Chronic liver disease	11 (8.3)
Charlson Comorbidity Index	2 (1–3)
Clinical status on ICU admission	
Mechanical ventilation	60 (45.1)
Vasopressor support	69 (51.9)
Laboratory findings	
WBC (/µL)	13,640 (5060–20,340)
Hemoglobin (g/dL)	10.1 (8.9–11.7)
Platelet (/µL)	136,000 (59,000–214,000)
Albumin (g/dL)	2.9 (2.6–3.2)
CRP (mg/dL)	12.5 (5.8–24.5)
Lactate (mg/dL)	3.0 (2.0–4.4)
PaO_2_/FiO_2_ ratio	216 (135–323)
Severity of illness	
SAPS 3 score	55 (47–63)
APACHE II score	24 (20–30)
SOFA score, initial	9 (7–11)
Outcome	
28-day mortality	22 (16.5)
In-hospital mortality	33 (24.8)

Data are presented as count (percentage) or median (interquartile range). Abbreviations: APACHE II, Acute Physiology and Chronic Health Evaluation II; COPD, chronic obstructive pulmonary disease; CRP, C-reactive protein; ICU, intensive care unit; PaO_2_/FiO_2_ ratio, ratio of arterial oxygen pressure to fractional inspired oxygen; SAPS 3, Simplified Acute Physiology Score 3; SOFA, Sequential Organ Failure Assessment.

**Table 2 ijms-24-09362-t002:** Top 10 upregulated and downregulated immune-related genes in patients with sepsis relative to healthy volunteers.

Top 10 Upregulated Genes			Top 10 Downregulated Genes		
Entrez ID	Gene Symbol	Fold Change	Entrez ID	Gene Symbol	Fold Change
6279	S100A8	16.17	973	CD79A	0.12
8876	VNN1	7.83	3120	HLA-DQB2	0.13
3148	HMGB2	7.23	122618	PLD4	0.14
6280	S100A9	5.43	1236	CCR7	0.15
301	ANXA1	5.42	259197	NCR3	0.17
665	BNIP3L	4.67	29802	VPREB3	0.18
200315	APOBEC3A	4.61	6932	TCF7	0.199
353514	LILRA5	4.49	933	CD22	0.201
1378	CR1	4.46	10578	GNLY	0.206
1604	CD55	4.29	974	CD79B	0.213

Abbreviations: S100A8, S100 calcium-binding protein A8; VNN1, vanin 1; HMGB2, high-mobility group box 2; S100A9, S100 calcium-binding protein A9; ANXA1, annexin A1; BNIP3L, BCL2 interacting protein 3-like; APOBEC3A, apolipoprotein B mRNA editing enzyme catalytic subunit 3A; LILRA5, leukocyte immunoglobulin-like receptor A5; CR1, complement C3b/C4b receptor 1; CD55, CD55 molecule; CD79A, CD79a molecule; HLA-DQB2, major histocompatibility complex, class II, DQ beta 2; PLD4, phospholipase D family member 4; CCR7, C-C motif chemokine receptor 7; NCR3, natural cytotoxicity triggering receptor 3; VPREB3, V-set pre-B cell surrogate chain 3; TCF7, transcription factor 7; CD22, CD22 molecule; GNLY, granulysin; CD79B, CD79b molecule.

**Table 3 ijms-24-09362-t003:** Comparisons of the expression levels of differentially expressed genes between patients with sepsis and healthy volunteers, and between patients who died and survived.

Patients with Sepsis vs. Healthy Volunteers *				
		Patients with Sepsis	Healthy Volunteers	*p*-Value
Upregulated genes	S100A8	19.8 (11.6–24.2)	7.6 (7.1–8.2)	<0.001
	S100A9	7.6 (3.8–10.3)	3.8 (2.6–4.2)	<0.001
	CR1	5.5 (2.5–8.8)	2.7 (2.3–2.8)	0.005
Downregulated genes	CD79A	0.3 (0.2–0.4)	0.5 (0.5–0.6)	<0.001
	HLA-DQB2	0.2 (0.2–0.2)	0.3 (0.3–0.4)	<0.001
	PLD4	0.5 (0.2–0.7)	0.9 (0.9–1.0)	<0.001
	CCR7	0.3 (0.1–0.4)	0.5 (0.5–0.6)	<0.001
**The Dead vs. Surviving among Patients with Sepsis ***				
		**Patients (Dead)**	**Patients (Surviving)**	***p*-Value**
Upregulated genes	S100A8	25.9 (24.1–27.8)	17.1 (11.4–21.2)	<0.001
	S100A9	10.7 (9.7–11.8)	6.3 (3.2–8.5)	<0.001
	CR1	10.3 (9.3–11.9)	4.0 (2.0–6.6)	<0.001
Downregulated genes	CD79A	0.5 (0.4–0.6)	0.2 (0.1–0.3)	<0.001
	HLA-DQB2	0.3 (0.2–0.3)	0.2 (0.1–0.2)	<0.001
	PLD4	0.7 (0.6–0.8)	0.4 (0.2–0.6)	<0.001
	CCR7	0.4 (0.4–0.5)	0.2 (0.1–0.3)	<0.001

Data are presented as median (interquartile range). * Values denote fold changes of genes.

**Table 4 ijms-24-09362-t004:** Receiver operating characteristic curve analysis and suggested optimal cut-off values of seven mRNAs in diagnosing sepsis and predicting in-hospital mortality among patients with sepsis.

mRNAs	Sepsis Diagnosis					
	AUC	SE	95% CI	Optimal Cut-Off	Sensitivity	Specificity
S100A8	0.931	0.026	0.880–0.982	≥9.0	94.0%	83.3%
S100A9	0.791	0.041	0.711–0.871	≥5.0	69.2%	100%
CR1	0.747	0.037	0.674–0.820	≥3.0	72.9%	100%
CD79A	0.113	0.028	0.059–0.168	NA	NA	NA
HLA-DQB2	0.003	0.003	0–0.009	NA	NA	NA
PLD4	0.013	0.013	0–0.038	NA	NA	NA
CCR7	0.016	0.010	0–0.036	NA	NA	NA
	**Mortality Prediction in Sepsis**					
	**AUC**	**SE**	**95% CI**	**Optimal** **Cut-Off**	**Sensitivity**	**Specificity**
S100A8	0.919	0.026	0.868–0.970	≥23.6	84.9%	88.0%
S100A9	0.863	0.033	0.797–0.928	≥9.6	78.8%	85.0%
CR1	0.966	0.014	0.939–0.993	≥9.0	84.9%	95.0%
CD79A	0.961	0.016	0.930–0.992	≥0.42	93.9%	92.0%
HLA-DQB2	0.936	0.021	0.895–0.978	≥0.23	84.9%	88.0%
PLD4	0.918	0.026	0.866–0.969	≥0.64	84.9%	88.0%
CCR7	0.918	0.026	0.866–0.969	≥0.39	78.8%	90.0%

Abbreviations: AUC, area under curve; SE, standard error; CI, confidence interval; NA, not applicable.

## Data Availability

The data discussed in this publication have been deposited in NCBI’s Gene Expression Omnibus and are accessible through GEO Series accession number GSE232753 (https://www.ncbi.nlm.nih.gov/geo/query/acc.cgi?acc=GSE232753).

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
