# Peer review of "Bioinformatics Analysis of Gene Expression Profiles for Diagnosing Sepsis and Risk Prediction in Patients with Sepsis"

_ijms, 2023, doi:10.3390/ijms24119362_

Round 1

Reviewer 1 Report

The publication deals with an interesting issue, namely the comparison of the blood transcriptome of sepsis patients versus healthy individuals. The authors identified several significantly regulated genes that could serve as sensitive markers for future diagostics and prognosis of the disease. 

The topic is very important, however, in its current version the paper is not suitable for publication.

specific comments. 

1 There are many publications that have studied the profile of gene expression changes both in whole blood and in individual fractions or using single cell RNAseq. The authors do not refer to these studies at all. In my opinion, the whole discussion should be rewritten and refer to papers in which they performed similar determinations. I wish also that the authors performed comparative analyses to other studies and presented them in the relevant chapter of the paper

I give only a few examples of papers that you should discuss and compare your results to published results:

1)     Blood transcriptome analysis of patients with uncomplicated bacterial infection and sepsis

2)     Bioinformatics Analysis for Multiple Gene Expression Profiles in Sepsis

3)     Bioinformatics Analysis of Gene Expression Profiles for Risk Prediction in Patients with Septic Shock

4)     Comprehensive Transcriptome Profiling of Peripheral Blood Mononuclear Cells from Patients with Sepsis

5)     Using RNA-Seq to Investigate Immune-Metabolism Features in Immunocompromised Patients With Sepsis

And much more 

2 There is no data on the number of patients (N) whose biological material was subjected to RNAseq and QPCR analysis

3 It is not clear on which material RNAseq was performed. The authors write about plasma aliquots - surely the experiment was performed on plasma RNA? Or was it perhaps performed on WBC or some isolated population? The sample collection section itself raises a concern.

4 The analysis of results should be expanded to include PCA, volcano plot analysis. 

5 The data analysis chapter should be strongly improved - there are strong generalizations, no information on differential analysis of gene expression

6 Raw RNAseq data needs to be included in the GEO repository

7 In Identification of DEG chapter, log should be corrected to log2

8 No adjusted p value information, also no description of what multiple testing correction was applied in the paper. DEGs should be determined based on adj.p val <0.05 and not on p without adjustment

9 QPCR expression should be counted as 2^ -delta delta Ct, not 2^ delta Ct

10 patient characteristics - lack data on blood characteristics

11 fig 1 A - poorly readable. Suggests to refer DEGs to GO DIRECT database from DAVID because it gives more precise ontology groups

12 fig 12 - change background color to white and fonts to black, also improve quality

13 Fig 3 and Fig 4 look like they were generated in Metascape - lack of any information and Metascape citations in materials and methods 

14 Tables 2 - is the ID an Entrez ID? Suggest adding the full name of the genes 

15 Tables 4 - sensitivity and specificity should be reported in ROC analysis

Author Response

## Response to Reviewer 1

General comment. The publication deals with an interesting issue, namely the comparison of the blood transcriptome of sepsis patients versus healthy individuals. The authors identified several significantly regulated genes that could serve as sensitive markers for future diagostics and prognosis of the disease. 

The topic is very important, however, in its current version the paper is not suitable for publication.

Response. We appreciate the reviewer’s helpful comments, which have substantially improved the quality of our manuscript. We are submitting a revised manuscript to address the reviewer’s concerns. Detailed point-by-point responses to these concerns are provided.

Specific comments. 

Comment 1 (C1). There are many publications that have studied the profile of gene expression changes both in whole blood and in individual fractions or using single cell RNAseq. The authors do not refer to these studies at all. In my opinion, the whole discussion should be rewritten and refer to papers in which they performed similar determinations. I wish also that the authors performed comparative analyses to other studies and presented them in the relevant chapter of the paper.

I give only a few examples of papers that you should discuss and compare your results to published results: 

1)     Blood transcriptome analysis of patients with uncomplicated bacterial infection and sepsis

2)     Bioinformatics Analysis for Multiple Gene Expression Profiles in Sepsis

3)     Bioinformatics Analysis of Gene Expression Profiles for Risk Prediction in Patients with Septic Shock

4)     Comprehensive Transcriptome Profiling of Peripheral Blood Mononuclear Cells from Patients with Sepsis

5)     Using RNA-Seq to Investigate Immune-Metabolism Features in Immunocompromised Patients With Sepsis And much more 

Response 1 (R1). Thank you for bringing our attention to this point, which we have not fully acknowledged in the original manuscript. As recommended, we have more incorporated previous publications and rewritten the Discussion section of the revised manuscript (pages 11–12, lines 399–475).

“Immune-related pathways were the most highly enriched biological pathways in patients with sepsis relative to healthy volunteers; this finding is consistent with previous bioinformatics studies, which also revealed DEGs were significantly enriched in pathways related to neutrophil activation, TNF signaling pathway, and cytokine secretion. These results suggest that the pathophysiology of sepsis is driven by “an aberrant or dysregulated host response to infection,” which is also reflected in the current definition of sepsis. Furthermore, a recent RNA sequencing study provided more insights into the pathophysiology of sepsis. The study compared RNA sequencing results between 18 immunocompromised patients with sepsis and 18 sequential organ failure assessment score-matched immunocompetent controls, and demonstrated that patients with sepsis were more likely to show compromised T cell function, decreased T cell diversity and altered metabolic signaling than controls. In this way, bioinformatics studies will increasingly contribute to unveiling the pathophysiology of sepsis.

Notably, the key genes upregulated during sepsis are responsible for regulating cell cycle progression and differentiation (i.e., S100A8 and S100A9) and immune responses (i.e., ANXA1, APOBEC3A, LILRA5, CR1, and CD55). In agreement to our findings, one Chinese prospective cohort study also performed a comprehensive transcriptome profile analysis and qPCR validation, and suggested S100A8, S100A9 and ANXA3 as key genes differentially expressed between sepsis patients and healthy controls. Potential underlying mechanisms of upregulated S100A8 and S100A9 may also include altering MyD88-dependent gene programs, which consequently prevent hyperinflammatory responses without impairing pathogen defense, and mediating endotoxin-induced cardiomyocyte dysfunction. Our qPCR results and ROC curve analyses confirmed the accuracy of S100A8, S100A9 and CR1 in diagnosing sepsis as well as in predicting in-hospital mortality among patients with sepsis. Thus, our results suggest that the expression levels of the key upregulated genes identified here may serve as potential biomarkers of sepsis.

In this study, the key genes downregulated during sepsis also encoded proteins responsible for the immune response, including the expression of the antigen-presenting cells, the regulation of cytokine production, and the activation of B and T lymphocytes (i.e., CD79A, HLA-DQB2, PLD4, and CCR7). This is in line with prior studies showing that genes involving Jak-STAT signaling, T cell receptor signaling, and natural killer cell-mediated pathways were downregulated. The results showed that genes participating in the immune response have mixed differential expression patterns of both up- and down-regulation. Considering sepsis has been found to manifest a balance between competing pro- and anti-inflammatory pathways, the downregulation of immune response genes in patients with sepsis implies the homeostatic regulation of immunity during sepsis. Thus, the failed homeostatic regulation of immunity may have consequently resulted in development and progression of sepsis. Interestingly, the key downregulated genes revealed an excellent accuracy in predicting in-hospital mortality of patients with sepsis, but failed to diagnose sepsis in the ROC curve analyses. Taken together, key downregulated genes helped us understand better about sepsis pathophysiology; however, they may not be useful as sepsis biomarkers compared with key upregulated genes.”

C2. There is no data on the number of patients (N) whose biological material was subjected to RNAseq and QPCR analysis

R2. Biologic material from all 133 patients with sepsis and 12 healthy volunteers was subjected to both RNAseq and qPCR. We have clarified this in the Results section of the revised manuscript (page 5, lines 199–200 and page 10, lines 303–305).

“This study initially identified 422 DEGs between 133 patients with sepsis and 12 healthy volunteers…”

“Next, we used qPCR to measure the expression levels of identified DEGs and thereby validate our bioinformatics analyses using molecular data in 133 patients with sepsis and 12 healthy volunteers.”

C3. It is not clear on which material RNAseq was performed. The authors write about plasma aliquots - surely the experiment was performed on plasma RNA? Or was it perhaps performed on WBC or some isolated population? The sample collection section itself raises a concern.

R3. Thank you for your comment. The subsection of Methods, “2.2 Sample collection”, was a part of our general protocol on storing blood samples, which might have led misunderstandings to the reviewer. Thus, we have clarified that this study had used 2 mL of whole blood for RNAseq in the Methods section of the revised manuscript (page 2, lines 78–86).

2.3. Total RNA isolation and quality analysis

For RNA isolation, the whole blood (2 mL) was also collected in in PAXgene tubes, using BD PAXgene blood RNA tubes (BD, cat. no. 762165). The total RNA was isolated from whole blood using the TRIzol reagent (Invitrogen, USA) following the manufacturer’s protocol. RNA quantity and purity were measured using a NanoDrop 2000 (Thermo Fisher Scientific, Wilmington, DE, USA). RNA quality, yield, and distribution were determined using an Agilent 2100 Bioanalyzer (Agilent Technologies, Santa Clara, CA, USA). Blood samples (5 mL) were also collected from healthy volunteers; these samples were also prepared using the method described above.”

C4. The analysis of results should be expanded to include PCA, volcano plot analysis. 

R4. We agree with the reviewer’s comment. As recommended, we have modified the Methods (page 3, lines 134–135) and Results (page 5, lines 200–203) sections and provided the PCA and volcano plot analyses as Figures 1A and 1B (page 5) in the revised manuscript.

Methods

“Principal component analysis (PCA) of the different groups’ samples was performed on the gene expression matrix.”

Results

“A PCA of global gene expression profiles revealed that sepsis patients were clearly separate from healthy volunteers (Figure 1A). In addition, distinct patterns of gene expression existed in sepsis patients when transcriptomic profiles of sepsis patients were compared to those of healthy volunteers (Figure 1B).”

C5. The data analysis chapter should be strongly improved - there are strong generalizations, no information on differential analysis of gene expression.

R5. Thank you for bringing our attention to this point, which we have not fully acknowledged in the original manuscript. We have reinforced information on differential analysis of gene expression in the Methods (page 3, lines 134–139) and Results (page 5, lines 200–203) sections of the revised manuscript.

Methods

“Principal component analysis (PCA) of the different groups’ samples was performed on the gene expression matrix. We set the threshold of differential expression to the default standard (i.e., |log2 (fold change [FC])| > 1 and adjusted P < 0.05) to identify significant DEGs between the two groups. Thus, significantly upregulated DEGs showed log2 FC > 1 and significantly downregulated DEGs showed log2 FC < 1. Significance was defined as an adjusted P value < 0.05 to control type I error in multiple tests.”

Results

“A PCA of global gene expression profiles revealed that sepsis patients were clearly separate from healthy volunteers (Figure 1A). In addition, distinct patterns of gene expression existed in sepsis patients when transcriptomic profiles of sepsis patients were compared to those of healthy volunteers (Figure 1B).”

C6. Raw RNAseq data needs to be included in the GEO repository.

R6. Thank you for your comment. However, uploading genetic information of the study population to an open-source database may be a sensitive issue. As we did not explain this issue to our study population while obtaining informed consent, we should discuss it with the Ethics Committee of our institution before including raw RNAseq data in the GEO repository. We hope the reviewer understands our situation.

C7. In Identification of DEG chapter, log should be corrected to log2.

R7. We have corrected the issue as recommended in the Methods section of the revised manuscript (page 3, lines 135–139).

“We set the threshold of differential expression to the default standard (i.e., |log2 (fold change [FC])| > 1 and adjusted P < 0.05) to identify significant DEGs between the two groups. Thus, significantly upregulated DEGs showed log2 FC > 1 and significantly downregulated DEGs showed log2 FC < 1. Significance was defined as an adjusted P value < 0.05 to control type I error in multiple tests.”

C8. No adjusted p value information, also no description of what multiple testing correction was applied in the paper. DEGs should be determined based on adj.p val <0.05 and not on p without adjustment.

R8. Thank you for the comment. We clarified that significance was defined as an adjusted P-value < 0.05 in the Methods section of the revised manuscript (page 3, lines 135–139).

“We set the threshold of differential expression to the default standard (i.e., |log2 (fold change [FC])| > 1 and adjusted P < 0.05) to identify significant DEGs between the two groups. Thus, significantly upregulated DEGs showed log2 FC > 1 and significantly downregulated DEGs showed log2 FC < 1. Significance was defined as an adjusted P value < 0.05 to control type I error in multiple tests.”

C9. QPCR expression should be counted as 2^ -delta delta Ct, not 2^ delta Ct

R9. Thank you for careful review of our manuscript. We have rephrased it in the Methods section of the revised manuscript (page 4, lines 167–168).

“The 2−ΔΔCT algorithm (ΔCT = Ct. target − Ct. reference) was employed for downstream data analysis.”

C10.  patient characteristics - lack data on blood characteristics.

R10. We have added more information on blood characteristics to Table 1 as recommended in the revised manuscript (page 4).

C11.  fig 1A - poorly readable. Suggests to refer DEGs to GO DIRECT database from DAVID because it gives more precise ontology groups

R11. As recommended, we have modified the figure, which is Figure 1C in the revised manuscript (page 6).

C12.  fig 12 - change background color to white and fonts to black, also improve quality

R12. As recommended, we have modified Figure 2 in the revised manuscript (page 7).

C13. Fig 3 and Fig 4 look like they were generated in Metascape - lack of any information and Metascape citations in materials and methods.

R13. We appreciate the reviewer’s careful reading of our manuscript. We have clarified the issue in the Methods section of the revised manuscript (page 3, lines 148–149).

“For GO functional annotation and KEGG pathway enrichment analyses, we used the WEB-based Gene SeT AnaLysis Toolkit (WebGestalt), web-based Database for Annotation, Visualization, and Integrated Discovery (DAVID) tool version 6.8, and Metascape”

C14. Tables 2 - is the ID an Entrez ID? Suggest adding the full name of the genes.

R14. We have provided Entrez IDs and added the full name of the genes as footnotes of Table 2 in the revised manuscript (page 9).

C15. Tables 4 - sensitivity and specificity should be reported in ROC analysis

R15. Thank you for bringing our attention to this point, which we have not fully acknowledged in the original manuscript. In addition to AUC values, we have modified the Methods section (page 4, lines 175­–176) and provided sensitivity and specificity of the optimal cutoff value of each mRNA in Table 4 (pages 10­–11) in the revised manuscript.

Methods

“The sensitivity and specificity were also calculated to suggest the optimal cutoff value of each gene.”

Reviewer 2 Report

Choi et al performed gene expression analysis of plasma samples from the diagnosis of sepsis. The study is interesting and well-designed. I have a few comments that the authors should address: 

Introduction: No expression studies shave been done on sepsis patient? Authors should present current evidence and identify gaps that they are addressing.

Results:

·       Were all patients at the same severity levels? If not, then it would be interesting to see DEGs between high and low-severity groups.

·       What is the y-axis in figure 1B? Also, A and B labels are missing

·       Can figure 2 be presented on white background

·       What values is table 3 showing? Fold change?

·       Weren’t all 10 up-and down-regulated genes tested in qPCR? Why?

Author Response

## Response to Reviewer 2

General comment. Choi et al performed gene expression analysis of plasma samples from the diagnosis of sepsis. The study is interesting and well-designed. I have a few comments that the authors should address: 

Response. We appreciate the reviewer’s words of encouragement and helpful comment. We are submitting a revised manuscript to address these concerns. Detailed point-by-point responses to these concerns are provided.

Specific comments. 

C1. Introduction: No expression studies shave been done on sepsis patient? Authors should present current evidence and identify gaps that they are addressing.

R1. We agree with the reviewer’s comment. Although there are many bioinformatics studies investigating patients with sepsis, genes suggested as potential biomarkers differed according to studies and some studies lacked experimental validation and information on clinical outcomes. In this regard, we have modified the Introduction section of the revised manuscript as recommended (page 2, lines 46­–52).

“In addition to the previous studies using datasets downloaded from public repositories, some prospective cohort studies also performed bioinformatics analysis to explore potential biomarkers for sepsis diagnosis. Although previous studies have provided important insight into biomarkers and pathophysiology of sepsis, more bioinformatics studies are warranted to validate the study results in association with real-world clinical outcomes.”

C2.  Results: Were all patients at the same severity levels? If not, then it would be interesting to see DEGs between high and low-severity groups.

R2. Thank you for the reviewer’s suggestion. We have regarded mortality as a major outcome of patients with sepsis in this study, which may denote the severity of sepsis. Although we appreciate the reviewer’s wonderful suggestion, an additional analysis according to the disease severity (i.e., SOFA score) would be beyond a scope of our current study.

C3.  What is the y-axis in figure 1B? Also, A and B labels are missing.

R3. We have clarified that Y-axis represents the number of genes and added labels in both figures. The two figures are Figures 1C–D in the revised manuscript (page 6).

C4.  Can figure 2 be presented on white background.

R4. Thank you for your comment. As recommended, Figure 2 is presented on white background in the revised manuscript (page 7).

C5.  What values is table 3 showing? Fold change?

R5. We have clarified that the values show fold changes as the footnote of Table 3 in the revised manuscript (page 10).

“*Values denote fold changes of genes.”

C6.  Weren’t all 10 up-and down-regulated genes tested in qPCR? Why?

R6. Thank you for bringing our attention to this point, which we have not fully acknowledged in the original manuscript. We initially performed qPCR in a whole study population comprising 133 patients with sepsis and 12 healthy volunteers. However, among the 10 upregulated and 10 downregulated genes, three upregulated and four downregulated genes showed consistent results with bioinformatics analysis. Thus, we provided qPCR results of the seven genes in Table 3 and proceeded with further analysis with them in Table 4. To address your concern, we have clarified this in the Results section of the revised manuscript (page 10, lines 303–307).

“Next, we used qPCR to measure the expression levels of identified DEGs and thereby validate our bioinformatics analyses using molecular data in 133 patients with sepsis and 12 healthy volunteers. Among the 10 upregulated and 10 downregulated genes, three upregulated and four downregulated genes showed consistent results with bioinformatics analysis.”

Round 2

Reviewer 1 Report

Thank you for the information. Now the article seems to be more accurate. I don't fully accept the explanation for the inability to deposit RNAseq results in the database. Based on these results, we are unable to identify specific individuals, so I maintain my opinion that such results should be publicly available

Why were the patients in the PCA presented in different colors? There is no explanation in the figure description.

Author Response

## Response to Reviewer 1

General comment. Thank you for the information. Now the article seems to be more accurate.

Response. We appreciate the reviewer’s helpful comments again, which have substantially improved the quality of our manuscript. We are submitting a revised manuscript to address the reviewer’s concerns. Detailed point-by-point responses to these concerns are provided.

Specific comments. 

Comment 1 (C1). I don't fully accept the explanation for the inability to deposit RNAseq results in the database. Based on these results, we are unable to identify specific individuals, so I maintain my opinion that such results should be publicly available.

Response 1 (R1). Thank you for bringing our attention to this point, which we have not fully acknowledged in the original manuscript. We deposited our RNA-Seq data in NCBI's Gene Expression Omnibus and are accessible through GEO Series accession number GSE232753 (https://www.ncbi.nlm.nih.gov/geo/query/acc.cgi?acc=GSE232753). We added this information in the revised manuscript.

C2. Why were the patients in the PCA presented in different colors? There is no explanation in the figure description.

R2. Thank you for careful review of our manuscript. We apologize for our carelessness and applied the same color to each group.

Reviewer 2 Report

The authors have satisfactorily addressed my comments. The paper can now be accepted. 

Author Response

## Response to Reviewer 2

C1. The authors have satisfactorily addressed my comments. The paper can now be accepted.

R1. We appreciate the reviewer’s helpful comments again, which have substantially improved the quality of our manuscript.